# Possible Link between Higher Transmissibility of Alpha, Kappa and Delta Variants of SARS-CoV-2 and Increased Structural Stability of Its Spike Protein and hACE2 Affinity

**DOI:** 10.3390/ijms22179131

**Published:** 2021-08-24

**Authors:** Vipul Kumar, Jasdeep Singh, Seyed E. Hasnain, Durai Sundar

**Affiliations:** 1Department of Biochemical Engineering and Biotechnology, Indian Institute of Technology (IIT) Delhi, New Delhi 110016, India; vipul.kumar@dbeb.iitd.ac.in (V.K.); jasdeep002@gmail.com (J.S.); 2Department of Life Science, School of Basic Sciences and Research, Sharda University, Greater Noida 201301, India

**Keywords:** B.1.1.7, B.1.617.2, COVID-19, E484Q, T478K and L452R mutation, N501Y mutation, spike protein

## Abstract

The Severe Acute Respiratory Syndrome Coronavirus 2 (SARS-CoV-2) outbreak in December 2019 has caused a global pandemic. The rapid mutation rate in the virus has created alarming situations worldwide and is being attributed to the false negativity in RT-PCR tests. It has also increased the chances of reinfection and immune escape. Recently various lineages namely, B.1.1.7 (Alpha), B.1.617.1 (Kappa), B.1.617.2 (Delta) and B.1.617.3 have caused rapid infection around the globe. To understand the biophysical perspective, we have performed molecular dynamic simulations of four different spikes (receptor binding domain)-hACE2 complexes, namely wildtype (WT), Alpha variant (N501Y spike mutant), Kappa (L452R, E484Q) and Delta (L452R, T478K), and compared their dynamics, binding energy and molecular interactions. Our results show that mutation has caused significant increase in the binding energy between the spike and hACE2 in Alpha and Kappa variants. In the case of Kappa and Delta variants, the mutations at L452R, T478K and E484Q increased the stability and intra-chain interactions in the spike protein, which may change the interaction ability of neutralizing antibodies to these spike variants. Further, we found that the Alpha variant had increased hydrogen interaction with Lys353 of hACE2 and more binding affinity in comparison to WT. The current study provides the biophysical basis for understanding the molecular mechanism and rationale behind the increase in the transmissivity and infectivity of the mutants compared to wild-type SARS-CoV-2.

## 1. Introduction

The Severe Acute Respiratory Syndrome—Coronavirus-2 (SARS-CoV-2), first detected in December 2019 in the Wuhan province of China, has caused the COVID-19 pandemic. As of August 18, 2021, there are more than 208,470,375 confirmed cases, and 4,377,979 people have lost their lives (https://covid19.who.int/) (accessed on 18 August 2021). The SARS-CoV-2 belongs to the family of beta corona virus, the same class of viruses responsible for previous pandemics caused by SARS-CoV and MERS [1,2,3]. SARS-CoV-2 possesses a large single-stranded RNA as genetic material and has four main structural components, namely, Envelope protein, spike protein, membrane protein and nucleocapsid [4,5,6]. The main structural element that enables this virus to attach to the host receptor is the spike glycoprotein, and it also gives the crown-like appearance to the virus, hence it is named Coronavirus [7,8,9]. The spike glycoprotein of SARS-CoV-2 attaches to the human angiotensin converting enzyme (hACE2) receptor and is then activated by another human enzyme, transmembrane protease serine (TMPRSS2), to enter the host cells [9,10,11]. Since spike is the primary target receptor for the entry and the main virulence factor of the virus, various therapeutic drugs and vaccines are being made and tested against it [11,12]. Although multiple medications such as remdesivir or hydroxychloroquine, lopinavir and ritonavir have been recommended by the World Health Organization (WHO) against COVID-19, their efficacy is still the topic of debate [13,14,15]. Similarly, WHO has issued an emergency use listing for certain vaccines such as BNT162B2 from Pfizer, AstraZeneca/Oxford COVID-19 vaccine, manufactured by the Serum Institute of India and SKBio, and Ad26.COV2.S, developed by Janssen (Johnson & Johnson) (https://www.who.int/covid-19/vaccines) (accessed on 20 April 2021). However, the SARS-CoV-2 cases are still increasing at an alarming rate all over the globe, and the primary rationale behind it is the rapid accumulation of mutations in the SARS-CoV-2.

In the past few months, multiple variants of the SARS-CoV-2 have been reported. Some of them are the variant of concern (VOC), which have increased the infectivity or have the potential of immune escape. Almost all the VOCs reported till now have mutations in the spike glycoprotein of the virus, which has increased the binding affinity of the virus to hACE2 or has conferred immune escape potential [16,17]. The Lineage B.1.1.7 or 20I/501Y.V1 (Alpha variant) was detected in the United Kingdom in September 2020. This variant increased the transmissibility by 40–80% and has been partially correlated with N501Y mutation in the receptor binding domain (RBD) of spike protein [18] (Figure 1A). In October 2020, B.1.351 (Beta variant) was detected in the South African population, which could infect more younger people and had three primary mutations in the RBD of spike protein, namely, N501Y, K417N and E484K [19,20]. Similarly, the lineage P.1 (Gamma variant) detected in January 2021 in the Brazilian population had three mutations of concern in spike RBD, namely, N501Y, K417T and E484K [17,21]. In our previous study, we had reported that N501Y mutation could enhance the ACE2 affinity and possibly confer resistance towards the antibodies [17]. Our results also indicated the reinfection potential of P1 and N501Y.V2 variants. In another study, it has been reported that N501Y mutation increases (dissociation constant: 22 nM to 0.44 nM) the binding affinity with hACE2 [22]. In India, lineage B.1.617 and B.1.618 have been recently reported, which had caused a rapid increase in the COVID-19 cases in the country [23,24]. The B.1.617 lineage, has been further divided into three sub lineages namely, B.1.617.1 (Kappa), B.1.617.2 (Delta variant) and B.1.617.3 [25] (Figure 1B). Out of these three sub lineages of B.1.617, the Delta variant has been identified as a variant of concern (VOC) and reported to be the main variant behind the second wave in India by WHO (https://www.who.int/en/activities/tracking-SARS-CoV-2-variants/) (accessed on 6 June 2021). The Kappa is characterized by E154K, L452R, E484Q, D614G, P681R, Q1071H mutations in the spike protein and Delta by T19R, L452R, T478K, D614G, P681R, D950N mutations while the B.1.617.3 lineage has T19R, L452R, E484Q, D614G, P681R mutations in the spike protein. All these lineages have conserved L452R, D614G and P681R (https://www.who.int/en/activities/tracking-SARS-CoV-2-variants/) (accessed on 6 June 2021). While in this study we have focused on the mutations within the RBD of spike protein, the D614G mutation (present outside the RBD region) has already been reported to increase the binding affinity with hACE2 and is susceptible to neutralization by antibodies [26].

While this paper was ready for submission a new sub variant of B.1.617 was detected and named as Delta^Plus^. It contains same mutations as Delta variant and two other mutations—K417N and W258L in the spike glycoprotein [27]. Further, the B.1.618 (triple mutant), recently detected in the four Indian states (Maharashtra, Delhi, West Bengal and Chhattisgarh), has been characterized by the deletion of Tyr145 and His146 as well as E484K and D614G mutation in the spike protein (https://cov-lineages.org/) (accessed on 10 July 2021) [24]. The sudden increase in COVID-19 cases in India is attributed to the Delta variant and its higher binding affinity towards hACE2 along with its immune escape ability [17,28]. In previous epidemiological and genomics study the sudden increase in the incidence of B.1.617.2 during February to April 2021 in India has been shown as the reason for the increased COVID-19 positivity rate [29]. A recent study focused on Delhi population sera survey, has reported that prevalence of B.1.617 lineage increased from 5 % in February to 60 % in April 2021 [29]. The loss of E484Q mutation and gain of T478K in the B.1.617.2 lineage directly correlated with increase in the positivity rate [29]. In another recent study, it has been reported that infection with B.1.617.2 variant could be controlled by antibodies induced due to prior infection or BNT162b2 vaccination, but with lower efficacy than the B.1.351 variant. This study further demonstrated that B.1.617.2 variant has greater lung cell entry and cell to cell fusion, indicating its higher lung infection capacity [30]. Although various studies have shown the phenotypic effect of the mutations, and increased transmissibility, limited data exist on comparative dynamics, molecular interactions, and changes in energetics due to these crucial mutations in the RBD domain of the spike protein of various mutants.

In the present study, we have aimed to investigate the thermodynamic effects of the mutations in the RBD region of the spike glycoprotein interacting with hACE2 and compare that with the wildtype. We accordingly studied two crucial variants Alpha, Kappa and Delta, which caused an increase in COVID-19 cases in various countries, including India. As in lineage B.1.617, Kappa and B.1.617.3 have same L452R and E484Q mutation in RBD of spike, while Delta has L452R and T478K, we have only considered Kappa and Delta in this study. These three variants (Alpha, Kappa and Delta) possess significant mutations in the RBD domain of the spike glycoprotein and have a higher infectivity rate. Therefore, to study and compare the dynamics, interactions and binding free energy of wildtype and spike protein variants with hACE-2 at the molecular level, we have performed the classical molecular dynamic (MD) simulations. 

## 2. Results

### The Mutant Spike Proteins Have a Better Binding Affinity with hACE2 in Comparison to Wildtype

The wildtype (WT) spike-hACE2 complex, along with the prepared and equilibrated Alpha, Kappa and Delta spike variants, were simulated for 200 ns. All the four structure complexes were first analysed for investigating the dynamics. In RMSD analysis, we found that the three complexes had a similar deviation around 2.5 Å from the initial structure over the 200 ns of simulations; Kappa_Spike-hACE2 (2.36 ± 0.27 Å), Alpha_Spike-hACE2 (2.62 ± 0.67 Å) and WT_Spike-ACE2 (2.82 ± 0.84 Å), while more around 3 Å deviation was found in the Delta_ Spike-hACE2 (3.12 ± 0.69 Å), as shown in Figure 2A. When RMSF of the simulated complexes was analysed, it was found that the residues number Arg355 to Phe400 of the spike protein was more flexible, especially in Delta. In addition, the fluctuation in the mutant residues was not high in the case of Kappa, although, it was found that residue Val445 had more fluctuations than the WT and N501Y mutants (Figure 2B). The average RMSF for WT was 2.95 ± 0.86 Å, for B.1.617 it was 2.66 ± 0.94 Å, for Delta it was 4.44 ± 1.62 Å, and for the Alpha spike protein it was 3.02 ± 1.09 Å. Though the RMSD and RMSF analysis suggested lesser stability of Delta in comparison to other studied complexes, no significant higher fluctuation was seen in the mutated residues in comparison to its overall structure. After analysing the fluctuation and deviations in the structures, the number of hydrogen bond count was calculated between the spike protein and hACE2 for all three structures. It was found that WT (12.23 ± 2.58) and Kappa (11.81 ± 2.07) had similar number of hydrogen bonds, followed by Delta (9.78 ± 2.40) and Alpha (9.19 ± 1.81) (Figure 2C). We further analysed the significant residues, to find out which of them has greater than 30% of the occupancy of hydrogen bond throughout the simulation. It was found that Alpha and Kappa spike mutants had more residues interaction with hACE2 than WT and Delta. In the case of WT and Delta, three residues (Tyr453, Thr500 and Gly502) and (Lys417, Gln493 and Gly502) of the spike protein were making a hydrogen bond with hACE2 for more than 30% of the simulation time, respectively. In comparison, in the Alpha spike mutant, five residues (Lys417, Ala475, Asn487, Thr500 and Gly502) were involved, and in Kappa, there were six residues (Lys417, Tyr449, Asn487, Tyr489, Tht500 and Gly502) of the spike protein that had significant hydrogen bond interactions with hACE2 (Appendix A). When the hydrogen bond interaction of mutated residues was checked, it was found that in Kappa, Q484 had only 0.1 fraction time interaction with E75 of hACE2 and in Delta, T478 had 1.5 fraction of time of interaction with Q353 of hACE2 throughout the simulations. Similarly, in case of Alpha, N501 had only 1.5 fraction of time of interaction with K353 of hACE2. Hence, the hydrogen bond analysis suggested that this mutation did not have any direct significant in terms of interaction with hACE2. It was observed that Gly502 was the critical residue interacting significantly with hACE2 in all four complexes. None of the mutated residues in Alpha, Kappa and Delta were found to be making significant hydrogen bonding with hACE2. Hence, it was essential to investigate if these mutated residues of the spike protein had interaction with any other spike residues or other interactions with hACE2 for any fraction of time. To analyse the changes in the interaction due to mutation, we extracted the three structures at the 50 ns interval from all the three simulated complexes. It was found that in the case of Kappa variant, in the 50th ns frame, neither Arg452 nor Gln484 were involved in any polar contact with other residues, while in 100th ns and 150th ns frame, it was found that Gln484 was making hydrogen bond contact with Ser349 and Asn450 of the spike protein itself, while in WT spike protein, Glu484 was making a hydrogen bond only with Ser349. It was observed that there was an increase in intra-chain interaction Spike protein due to mutation of E484Q. In Delta, no major interactions of mutated residues were found in comparison to WT spike, however in 100th ns frame, Lys478 was making intra-chain interaction with Ser476 of spike that was not found in case of Thr478 of WT spike. Similarly, when the Alpha variant was compared with WT, it was found that due to Asn to Tyr mutation at 501st residue, there was an increase in the hydrogen bonding with Lys353 of hACE2 (Figure 3).

The increase in the intra-chain interaction in case of B.1.617 indicated that it may interfere in the human antibodies’ interaction with the spike protein. In the case of Alpha, the increase in hydrogen bond contact with hACE2 indicated higher binding affinity of this mutant with hACE2 in comparison to WT. The MM/GBSA binding free energy has been earlier reported to correlate with the binding affinity between the complexes [31,32]. However, it is mainly used for comparing the binding energies of the studied complexes, not for absolute free energy calculations. Therefore, to assess and compare the binding affinity of the spike protein towards hACE2, we calculated the MM/GBSA binding free energy by extracting twenty structures in equal spans from 50th to 200 ns of the simulated trajectories. It was found that Alpha (−103.35 ± 16.31 kcal/mol) and Kappa (−101.90 ± 18.40 kcal/mol) spike proteins had a similar and higher binding affinity with hACE2 in comparison to WT (−96.87 ± 14.57 kcal/mol) (Figure 2D). Surprisingly, the MM/GBSA binding free energy of Delta with hACE2 was far less (−37.03 ± 22.79 kcal/mol) in comparison to all the studied complexes. Further, to calculate the energy contribution of individual mutated residues, prime energy was calculated for the twenty extracted structures, which showed that Kappa and Delta had a more stabilizing effect on the spike protein compared to WT, a recent study also shows similar results [33]. The average energy contribution of Arg (−50.90 ± 3.99 kcal/mol) in comparison to Leu (−22.15 ± 2.60 kcal/mol) at 452nd position of the spike protein was found to be high. The energy contribution of Gln (−56.03 ± 2.31 kcal/mol) in comparison to Glu (−47.12 ± 2.25 kcal/mol) at 484th position of spike protein was relatively higher. Similarly, Lys (−9.40 ± 2.66 kcal/mol) was favourable than Thr (−3.20 ± 2.93 kcal/mol) at 478th position. However, in the case of the Alpha mutant, it was noticed that Asn (−64.31 ± 3.59 kcal/mol) at 501st position was more energetically favourable than Tyr (−31.55 ± 3.26 kcal/mol) (Table 1). 

Although these binding energy calculations are theoretical and cannot be taken as absolute values, however, they are typically used for the comparison of binding affinity of the complexes with respect to each other. The interactions and binding energy calculations showed that in B.1.617 variant, there is a decrease of energy due to mutations as well as change in intra-chain interactions, which may lead to stabilization and interference with neutralizing antibodies interactions.

Overall, a significant increase in the binding affinity was observed in case of Kappa and Alpha variant in comparison to WT. However, the MM/GBSA binding energy of Delta with hACE2 was less in comparison to WT, suggesting that there must be some other ways these spike RBD mutations of Delta variant are helping in increased transmission but not by increasing the affinity with hACE2. While the Delta and Kappa mutations were found to be stabilizing the spike protein, but not N501Y of Alpha, increase/change in the intrachain interaction in the spike protein was observed in all the studied variants. Therefore, it can be interpreted that stabilization of the spike protein, increase of binding energy and increase in intra-chain interactions are crucial and are somehow aiding the Kappa variant, whereas in the Delta variant, it is the stabilizing of spike and increases in the intra-chain interactions. In the Alpha spike mutant, increases in the hydrogen-bond interaction and binding affinity with hACE2 could be the reason for more transmissivity of this mutant.

## 3. Discussion

The recent variants of SARS-CoV-2 are cause of the second wave of the COVID-19 around the world and setback to healthcare infrastructure specially in India [28,34]. The transmission of the three variant of concerns (VOC), namely, Alpha, Beta and Delta identified in UK, South Africa and India, respectively were drivers of subsequent infection waves in these nations (https://www.who.int/en/activities/tracking-SARS-CoV-2-variants/) (accessed on 6 June 2021). Alpha, the first VOC initially discovered in September 2020 in the UK population has four main mutations (H69-, V70-, N501Y and D614G) in the spike protein [35]. These mutations are reported to be the mutations of concern, in other words, these mutations are positively selected by the virus for its higher transmission. These mutations were found in various other SARS-CoV-2 variants as well, which emerged after it. After Alpha, the second main VOC detected was Beta in October 2020 in the South African population, and it had five main mutations, which were reported to be beneficial for transmission—L18F, K417N, E484K, N501Y and D614G [35]. The N501Y and D614G was conserved in both Alpha and Beta VOC and are believed to be crucial mutations for their higher transmission and infectivity. In a recent study, where 12 monoclonal antibodies were tested for their neutralizing activity against Alpha and Beta variants, it was found that N501Y of Alpha variant modulated interaction of neutralizing antibodies only, while in case of Beta, complete loss of activity was observed in most of the antibodies, mediated by K417N and E484K, in comparison to wildtype [36]. The same study further reported that when convalescent plasma from the 20 patients infected before the emergence of Alpha was investigated, it lost >2.5-fold neutralizing activity against Beta, while maintaining activity against Alpha. Additionally, when the efficacy of Moderna and Pfizer vaccines were tested, it was found that there was no loss of neutralizing activity against Alpha, whereas every sample lost activity against Beta [36]. Another similar study, where convalescent sera from infected people and vaccine recipients were tested against Alpha, suggested that it is not a neutralization escape VOC in terms of vaccine efficacy. Several studies including the current one has indicated that N501Y mutation is the main reason behind the increase of Alpha transmission [22,37]. Overall, recent studies suggest that though the Alpha variant has higher transmission, it is not an escape variant and could be neutralized by the vaccines available and will be available in the near future [36]. The VOC next to Alpha, i.e., Beta has been found to be greater concern than Alpha in terms of its neutralization by convalescent plasma of the infected individuals and Moderna and Novavax vaccines [38].

Earlier this year, in March 2021, the B.1.617 lineage found in India transmitted rapidly and is being investigated for its role in severity and mortality [34]. Recently, sub lineages of the B.1.617 - Kappa, Delta and B.1.617.3 were reported and characterized. The B.1.617.1 is characterized by E154K, L452R, E484Q, D614G, P681R, Q1071H mutations in the spike protein. In a recent study on B.1.617 lineages, it has been shown that the P681R has highest impact in increasing the fusion activity, followed by E484K and L452R [39]. Further, when the Kappa spike mutant was tried to be neutralized with Pfizer vaccine sera, it was found that E484K conferred a ten-fold reduction in neutralisation, E484Q had a slightly milder yet significant impact, however, with E484Q and L452R combined, there was a statistically significant loss of sensitivity [39]. In another study, two-fold reduction in the neutralization efficacy of Covaxin vaccine (BBV152) was observed against B.1.617.2 variant [40]. Combining previously published literature with our current observations, the alpha and delta SARS-CoV-2 variants with their mutations have optimally struck balance between higher transmission and immune evasive capabilities. Overall, the previous studies reported against the B.1.617 variant have indicated a slight decrease of neutralizing activity of vaccines in comparison to wildtype, however, they still provided significant protection. Similarly, previous studies have reported that L452R, E484Q/K, P681R and T478K might have role in the increased transmissibility, while the molecular level rationale is not clear [39,40,41]. This was investigated in this study.

In the current study, we described the interactions of mutant spike RBD with hACE2 of wildtype, Alpha, Kappa and Delta variants. The binding affinity was found to be least in case of Delta, while Kappa and Alpha spike RBD had higher binding affinity with hACE2 in comparison to wildtype. The results of binding free energy calculations suggested that E484Q and N501Y mutations are crucial for increasing the binding affinity. The comparative MM/GBSA binding energy calculations of N501Y reported here positively correlate with the available experimental absolute binding free energy reported elsewhere [22]. Further, it was found that the L45R, E484Q and T478K mutations are highly energetically favourable for the spike protein based on the prime energy calculations of the mutated residues. Though the mutations do not change, the molecular interactions between the hACE2 and spike significantly, the snapshots from the MD simulations clearly indicated the change and increase of the intra-chain interactions in the mutated spike proteins, possibly interfering with the neutralising antibodies. Further analysis of these mutants with neutralizing antibodies is expected to provide more mechanistic insights.

## 4. Materials and Methods

### 4.1. MD Simulations

The X-ray crystal structure of SARS-CoV-2, spike RBD bound with hACE2 was retrieved from Protein Data Bank (PDB) having PDB ID 6M0J. Along with wildtype, three mutants of the spike protein were created, namely, Alpha (N501Y), Kappa (L452R and E484Q) and Delta (L45R and T478K) using the Maestro Suite of Schrodinger software (2020-3, NY, USA) [31]. All the four structures were then pre-processed for missing side chains, deleting waters, the addition of hydrogens, hydrogen bond optimization and restrained minimization using the protein preparation wizard of Schrodinger software (2020-3, NY, USA) [31]. The prepared mutated structures were then subjected to classical molecular dynamics for 50 ns for the stabilization of the mutated structures and the last frame structure was taken for further studies. The following protocol was adopted for the MD simulations of all four prepared structures—each system was solvated with the TIP3P water model in an orthorhombic periodic boundary box. To prevent interaction of the protein complex with its own periodic image, the distance between the complex and the box wall was kept at 10 Å. The system was then neutralized by the addition of appropriate number of Na^+^/Cl^−^ ions depending on the complex using OPLS3e forcefield. Then the energy of the prepared systems was minimized by running 100 ps low-temperature (10 K) Brownian motion MD simulation (NVT ensemble) to remove steric clashes and move the system away from an unfavourable high-energy conformation. Further, the minimized systems were equilibrated in NVT and NPT ensembles using the “relax model system before simulation” option in the Desmond Schrodinger suite [31]. The equilibrated systems were then subjected to 200 ns unrestrained MD simulations in NPT ensemble with 300 K temperature maintained by Nose–Hoover chain thermostat constant pressure of 1 atm maintained by Martyna–Tobias–Kelin barostatand an integration time step of 2 fs with a recording interval of 200 ps.

### 4.2. Analysis of the MD Simulation

The root mean square deviation (RMSD), root mean square fluctuation (RMSF), number of hydrogen bonding was calculated using the simulation event analysis tool of the Desmond Suite integrated into Schrodinger software. Further, the occupancy of the hydrogen bonding between the spike protein and hACE2 was calculated using visual molecular dynamics (VMD) (1.9.4, UIUC, Champaign, IL, USA) [42]. The molecular mechanics generalized born surface area (MM/GBSA) free binding energy between spike proteins and hACE2 was calculated using the prime module of Schrodinger software [31]. Twenty structures extracted from 50 ns to 200 ns from each of the trajectories were used for this computation using the following equation:MMGBSAΔGbind=ΔGcomplex−ΔGreceptor+ ΔGligand
$$\Delta G={\Delta E}_{\mathrm{gas}}+{\Delta G}_{\mathrm{sol}}-{T\Delta S}_{\mathrm{gas}}$$
$${\Delta E}_{\mathrm{gas}}={\Delta E}_{\mathrm{int}}+{\Delta E}_{\mathrm{ele}}+{\Delta E}_{\mathrm{vdw}}$$
$${\Delta G}_{\mathrm{sol}}={\Delta G}_{\mathrm{gb}}+{\Delta G}_{\mathrm{surf}}$$

The binding free energy (ΔG_bind_) is dissociated into binding free energy of the complex, spike and hACE2. The gas–phase interaction energy (ΔE_gas_) was calculated as the sum of electrostatic (ΔE_elec_) and Van der Waal (ΔE_vdw_) interaction energies, while internal energy was neglected. The solvation free energy (ΔG_sol_) contains non-polar (ΔG_surf_) and polar solvation energy (ΔG_gb_), which was calculated by using the VSGB solvation model and OPL3e force field, while the entropy term was neglected by default [31,43].

The energy contribution of the mutated residues was then compared with wildtype residues. The Prime module of Schrodinger software (2020-3, NY, USA) was used for calculation of the energy contribution of the individual residues. The solvent model used here was surface generalized born (SGB), with variable dielectric enabled, the internal dielectric was 1.00 and solvent dielectric was 80.00 [31].

The following equation was used for the calculation of prime energy of individual residues:Total energy=covalent total + non−bonded total + other − SGB14|SGB − torsinol

Here, other energy=SGB self, nonpolar, hydrogen bond, packing, self − contact.

## 5. Conclusions

In this study, MD simulations were performed to compare the binding energy, interactions and change in dynamics of spike (RBD)–hACE2 complexes, namely WT, Alpha, Kappa and Delta. It has been shown that mutants have a higher number of significant hydrogen bond interactions with hACE2, and the binding free energy of the mutants is also higher in comparison to WT, except in case of Delta. In the B.1.617 lineage, the mutations were favourable in terms of making the spike energetically stable as well as in terms of intra–chain residue interactions. In alpha spike, the mutation led to an extra interaction and higher binding affinity with hACE2 compared to WT. The increased molecular level interaction dynamics of spike–hACE2 and the predicted increased structural stability of its spike protein and hACE2 affinity can be possibly linked to higher transmissibility of B.1.617 and Alpha variants of SARS-CoV-2. 

## Figures and Tables

**Figure 1 ijms-22-09131-f001:**
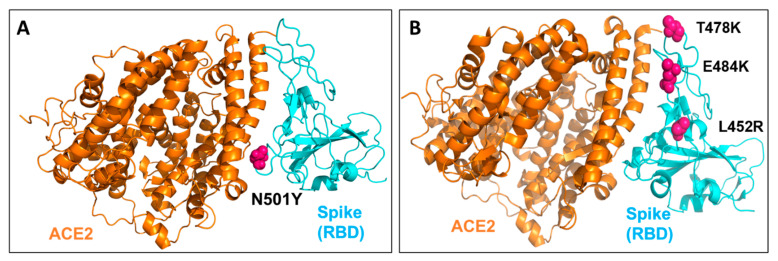
The structure of the receptor binding domain (RBD) of SARS-CoV-2 spike protein complexed with human angiotensin converting enzyme 2 (hACE2) receptor. (**A**) The sphere shape residues in hot pink colour show N501Y mutation in the spike protein of SARS-CoV-2. (**B**) At L452R, T478K and E484Q mutations in the spike protein (RBD) of B.1.617 lineage.

**Figure 2 ijms-22-09131-f002:**
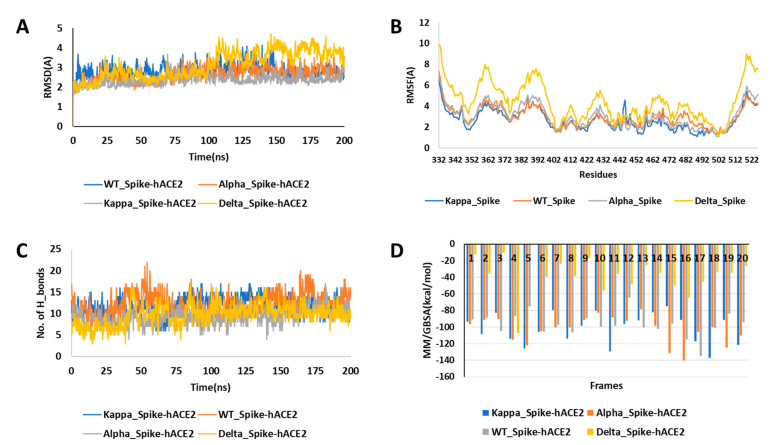
MD simulation analysis of the three simulated complexes. (**A**) RMSD plot showing similar deviation of all the simulated structures. (**B**) RMSF plot reveals that Residues 350-400 of the spike receptor binding domain (RBD) are more flexible, while the mutated residues have lesser fluctuation and are also comparable in all three structures. (**C**) The number of hydrogen bond count indicates that WT and Kappa variant have similar and higher number hydrogen bonds compared to Delta and Alpha variants. (**D**) MM/GBSA binding free energy of the 20 structure complexes extracted from each trajectory at equal span, suggesting that Kappa and Alpha spike variants have higher affinity for hACE2 in comparison to Delta and WT.

**Figure 3 ijms-22-09131-f003:**
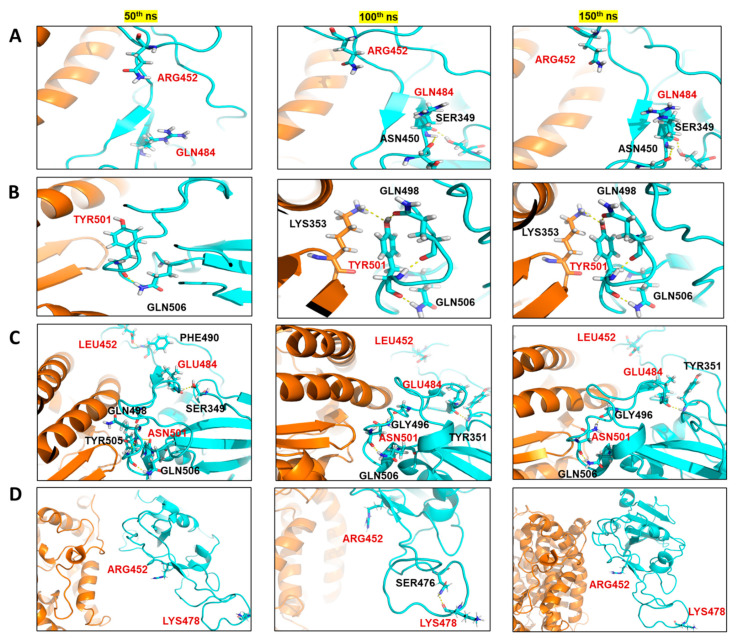
Comparing the interaction of the mutated residues and wild-type residues in the three structures extracted at 50 ns span from the simulated trajectories: spike protein (turquois color), and hACE2 (orange color). (**A**) Kappa spike variant and its interactions; Gln484 of the spike protein making intra-chain hydrogen bonding with Ser349 and Asn450 in the 100th and 150th ns frame. (**B**) Alpha spike variant interactions. The hydrogen bond interaction of Tyr501 of mutant spike protein with Lys353 of hACE2 in the 100th and 150th ns frame. (**C**) The interaction of wild type residues at 50th, 100th and 150th ns of the simulation shows that Glu484 of spike protein had only one hydrogen bond interaction with Ser349 or Tyr351 of spike itself. Similarly, Asn501 of spike was making hydrogen bond interactions with its residues only. (**D**) In the Delta spike variant, at 100th ns frame, an addition of a hydrogen bond of Lys478 with Ser476 was observed.

**Table 1 ijms-22-09131-t001:** Residue wise energy contribution of the mutated residues compared with the wildtype for the twenty structures extracted from 50 to 200 ns of the simulation for all the three complexes.

Kappa (kcal/mol)		Delta (kcal/mol)		WT (kcal/mol)		Alpha (kcal/mol)		
R452	Q484	R452	K478	L452	E484	N501	T478	Y501
−47.16	−51.43	−48.67	−7.51	−23.45	−47.06	3.44	3.44	−31.72
−50.69	−57.57	−49.32	−7.05	−23.04	−48.53	9.92	9.92	−27.6
−42.13	−56.15	−47.15	−10.92	−23.89	−45.52	3.81	3.81	−32.73
−52.43	−56.51	−45.49	−11.73	−20.05	−48.94	2.89	2.89	−31.96
−54.38	−58.47	−47.68	−9.31	−23.68	−50.2	2.03	2.03	−26.71
−54.05	−56.86	−46.76	−12.17	−25.88	−47.91	2.60	2.60	−29.47
−56.11	−53.98	−50.68	−7.42	−21.65	−47.86	4.45	4.45	−37.94
−47.14	−56.4	−45.18	−10.38	−22.44	−44.1	7.27	7.27	−33.03
−56.47	−57.21	−50.03	−13.46	−22.87	−43.55	−3.05	−3.05	−31.87
−54.79	−53.25	−48.51	−10.35	−22.05	−48.18	3.55	3.55	−33.2
−50.07	−51.51	−52.83	−6.85	−20.9	−46.46	7.14	7.14	−33.69
−45.38	−53.04	−55.77	−9.25	−17.67	−45.08	1.14	1.14	−25.84
−49.22	−59.17	−50.34	−11.01	−25.26	−49.82	4.00	4.00	−34.98
−51.75	−58.8	−47.65	−9.99	−16.19	−47.49	−0.80	−0.80	−29.93
−45.95	−55.64	−45.07	−14.89	−18.58	−47.46	2.51	2.51	−34.11
−48.85	−56.21	−45.52	−11.16	−20.55	−49.06	4.97	4.97	−34.51
−48.93	−56.69	−49.63	−6.93	−24.27	−46.24	5.18	5.18	−31.49
−54.01	−56.35	−48.91	−4.89	−23.78	−50.55	1.66	1.66	−32.64
−55.01	−59.25	−54.44	−7.28	−21.35	−41.95	0.17	0.17	−32.55
−53.5	−56.27	−53.66	−5.53	−25.55	−46.55	1.23	1.23	−25.06
**−50.90 ± 3.99**	**−56.03 ± 2.31**	**−49.16 ± 3.11**	**−9.40 ± 2.66**	**−22.15 ± 2.60**	**−47.12 ± 2.25**	**−64.31 ± 3.59**	**−3.20 ± 2.93**	**−31.55 ± 3.26**

## Data Availability

The authors confirm that the data supporting the findings of this study are available within the article and/or its Appendix A.

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
