# Peer review of "Possible Link between Higher Transmissibility of Alpha, Kappa and Delta Variants of SARS-CoV-2 and Increased Structural Stability of Its Spike Protein and hACE2 Affinity"

_ijms, 2021, doi:10.3390/ijms22179131_

Round 1
Reviewer 1 Report
The paper provides an explanation and a hypothesis on why variants of COVID virus are differently contagious. The paper is highly applicable, and the figures are very informative. The references are not many (could be improved) but this is not a major issue. I recommend acceptance of the paper after some references are added.
Author Response
Comment 1: The paper provides an explanation and a hypothesis on why variants of COVID virus are differently contagious. The paper is highly applicable, and the figures are very informative. The references are not many (could be improved) but this is not a major issue. I recommend acceptance of the paper after some references are added.
Response: We thank the reviewer for his comments. As suggested, relevant references (# 12, 20, 23, 24, 27, 33 and 34) have now been included in the revised manuscript.
Reviewer 2 Report
Dear authors
I Have appreciated your manuscript very much. All related experimental simulations are very interesting and detailed, even not so easy to read. The discussion is well conduct and commendable. The Abstract is very clear.
I’ve only few suggestions or requests before editing, as in following items:
Pag 2 line 39: in my opinion it’s better to update data about Covid 19 on the last disposable date, instead of June 15 2021.
Pag 3 line 107: You reported a new variant, but I didn't find any notice concerning B.1.1618 (triple mutant) variant in your suggested bibliography (https://cov-lineages.org). May you help me to find this.
Pag 6 line 212: Your results concerning Delta and Kappa lower binding free energy with hACE2 in comparison to Alfa are difficult to understand. These data are in contrast with the clinical experience, and with the published papers reporting Delta variant as the more infectious and life-threatening for humans. In effect, in page 8 line 248 you hypothesize in this variant some stabilizing of Spike protein and increasing in the intra-chain interactions. Could you better explain and validate this assumption by inserting data or scientific references if available.
Pag 9 line 303: better correct “B.617.2” with “B.1617.2
Finally, I would prefer putting “Materials and Methods” after the “Introduction”, as in the scientific paper’s classical structure.
Best regards
Author Response
Comment 1: Page 2 line 39: in my opinion it’s better to update data about Covid 19 on the last disposable date, instead of June 15 2021.
Response: We have now included most updated information on COVID 19 in the revised manuscript.
Comment 2: Page 3 line 107: You reported a new variant, but I didn't find any notice concerning B.1.1618 (triple mutant) variant in your suggested bibliography (https://cov-lineages.org). May you help me to find this.
Response: When this manuscript was prepared for submission, the B.1.1.618 (Triple mutant) cases were just rising in India. Almost all the reports on these were limited to only popular news articles. Since the Pango nomenclature (https://cov-lineages.org/) is being used by researchers and public health agencies worldwide to track the transmission and spread of SARS-CoV-2, we included this reference in the original manuscript. This website documents all the lineages and spread of the virus. In the revised manuscript, we have now provided the exact page link to find the lineage and its details (https://cov-lineages.org/lineage.html?lineage=B.1.618 ). We have also added one relevant scientific article reference (# 24) in the revised manuscript regarding the triple mutant.
Comment 3: Page 6 line 212: Your results concerning Delta and Kappa lower binding free energy with hACE2 in comparison to Alfa are difficult to understand. These data are in contrast with the clinical experience, and with the published papers reporting Delta variant as the more infectious and life-threatening for humans. In effect, in page 8 line 248 you hypothesize in this variant some stabilizing of Spike protein and increasing in the intra-chain interactions. Could you better explain and validate this assumption by inserting data or scientific references if available.
Response: We understand the reviewer’s concern that the predicted binding energies were slightly different from the experimental data. It is emphasized that these binding energies are not absolute and only show the comparison with each other. However, it is evident from the presented data that the delta variant has less binding affinity when compared to other variants studied. Based on the several structural interactions and mutant residue energies calculated from the stable molecular dynamics simulation trajectory, we have hypothesized that the mutations may be providing better stability in delta variant and the changes in the interactions could be the reason for the increase in the transmissibility. In line with our results and hypothesis, a recent peer-reviewed article shows a similar result related to the stabilizing effect of the delta variant mutations [1]. This reference (#33) has now been included in the revised manuscript.
Comment 4: Page 9 line 303: better correct “B.617.2” with “B.1617.2
Response: This typo has been corrected in the revised manuscript.
Comment 5: Finally, I would prefer putting “Materials and Methods” after the “Introduction”, as in the scientific paper’s classical structure.
Response: The structure of article is according to the format suggested by the Journal.
References
- Pascarella S, Ciccozzi M, Zella D, Bianchi M, Benedetti F, Benvenuto D, Broccolo F, Cauda R, Caruso A, Angeletti S, Giovanetti M, Cassone A. SARS-CoV-2 B.1.617 Indian variants: Are electrostatic potential changes responsible for a higher transmission rate? J Med Virol. 2021 Jul 14. doi: 10.1002/jmv.27210. Epub ahead of print. PMID: 34260088.